# Thymoquinone Inhibition of Chemokines in TNF-α-Induced Inflammatory and Metastatic Effects in Triple-Negative Breast Cancer Cells

**DOI:** 10.3390/ijms24129878

**Published:** 2023-06-08

**Authors:** Getinet M. Adinew, Samia Messeha, Equar Taka, Bereket Mochona, Kinfe K. Redda, Karam F. A. Soliman

**Affiliations:** 1Division of Pharmaceutical Sciences, College of Pharmacy and Pharmaceutical Sciences, Institute of Public Health, Florida A&M University, Tallahassee, FL 32307, USA; getinet1.mequanint@famu.edu (G.M.A.); samia.messeha@famu.edu (S.M.); equar.taka@famu.edu (E.T.); kinfe.redda@famu.edu (K.K.R.); 2Department of Chemistry, College of Science and Technology, Florida A&M University, Tallahassee, FL 32307, USA; bereket.mochona@famu.edu

**Keywords:** thymoquinone, triple-negative breast cancer, cytokine, chemokine, inflammation

## Abstract

The lack of identifiable molecular targets or biomarkers hinders the development of treatment options in triple-negative breast cancer (TNBC). However, natural products offer a promising alternative by targeting inflammatory chemokines in the tumor microenvironment (TME). Chemokines are crucial in promoting breast cancer growth and metastasis and correlate to the altered inflammatory process. In the present study, we evaluated the anti-inflammatory and antimetastatic effects of the natural product thymoquinone (TQ) on TNF-α-stimulated TNBC cells (MDA-MB-231 and MDA-MB-468) to study the cytotoxic, antiproliferative, anticolony, antimigratory, and antichemokine effects using enzyme-linked immunosorbent assays, quantitative real-time reverse transcription–polymerase chain reactions, and Western blots were used in sequence to validate the microarray results further. Four downregulated inflammatory cytokines were identified, CCL2 and CCL20 in MDA-MB-468 cells and CCL3 and CCL4 in MDA-MB-231 cells. Furthermore, when TNF-α-stimulated MDA-MB-231 cells were compared with MDA-MB-468 cells, the two cells were sensitive to TQ’s antichemokine and antimetastatic effect in preventing cell migration. It was concluded from this investigation that genetically different cell lines may respond to TQ differently, as TQ targets CCL3 and CCL4 in MDA-MB-231 cells and CCL2 and CCL20 in MDA-MB-468 cells. Therefore, the results indicate that TQ may be recommended as a component of the therapeutic strategy for TNBC treatment. These outcomes stem from the compound’s capacity to suppress the chemokine. Even though these findings support the usage of TQ as part of a therapy strategy for TNBC associated with the identified chemokine dysregulations, additional in vivo studies are needed to confirm these in vitro results.

## 1. Introduction

Breast cancer (BC) continues to be a significant global public health burden and the primary cause of cancer-related death in women [1]. Triple-negative breast cancer (TNBC), a very aggressive subtype of BC that accounts for 15–20% of breast cancer cases, has limited access to targeted therapy because of the lack of hormone receptors (ER and PR) and HER2 expression [2] in the tumor cells. There are various factors involved in the initiation and progression of TNBC. The tumor microenvironment (TME) influences the epigenetic and gene expression patterns that promote aggressive tumor phenotypes and therapy resistance by releasing different molecules [3,4]. In TNBC, the TME is associated with altered immune response, enhanced cell proliferation, migration, angiogenesis, and chemoresistance [5]. In addition, tumor macrophages, myeloid-derived suppressor cells, endothelial cells, and stromal fibroblasts release chemokine (CC) in the inflammatory TME [6].

Chemokines (CCs) are immunoregulatory proteins essential for immunity action and play a key role in inflammation, wound healing, angiogenesis, tumorigenesis, and cancer immunoediting [7]. In the last decade, CCs’ importance in breast cancer growth, angiogenesis, immune suppression, and site-directed metastasis has been well established [8]. The proliferation, migration, and differentiation of all cell types in the TME, including cancerous cells, fibroblasts, and endothelial cells, is regulated by inflammatory cells, including the CCs and cytokines [9].

Tumor cells control the expression of CCs to draw in inflammatory cells and use these elements to accelerate the development and spread of tumors [10], which implies that CCs and CC receptors in an inflammatory TME can support the proliferation of tumors. Tumor cells also divert inflammatory pathways, such as selectin–ligand interactions, matrix metalloproteinase (MMP) synthesis, and CC activities, later in the tumorigenic process to promote neoplastic dissemination and metastasis [5].

The inflammatory mediators induced by the inflammatory response can release tumor necrosis factor (TNF-α) and nonspecific proinflammatory cytokines, which can induce chemokine expression to promote inflammation [11]. TNF-α promotes tumor growth by causing inflammation, and it might be a target for cancer treatment [12]. The proinflammatory cytokine TNF-α also significantly mediates inflammatory agents in the TME. Moreover, TNF-α controls a cascade of events, involving cytokines, CCs, adhesions, MMPs, and angiogenesis [13].

Furthermore, the tumorigenic process is influenced by the cancer tissue-resident cells and recruited immune cells that express various chemokine receptors and ligands [14]. Chemokines are chemotactic cytokines that control cell orientation and trafficking, as well as their binding to glycosaminoglycans, which are crucial for holding the migration of immune cells and the development of lymphoid organs [15]. Meanwhile, chemokine dysregulation and the associated receptors are associated with various human diseases, including cancer [16]. In cancer, chemokines are crucial in cell migration and cell–cell signaling, significantly impacting tumor growth [14]. In addition to being shown to promote tumor growth, chemokines are also correlated to tumor progression and the establishment of tumor cells at distant organ sites [17]. Chemokines also regulate tumor cell stemness, proliferation, and invasiveness, as well as stromal cell neoangiogenesis, neurogenesis, and fibrogenesis [18].

The lack of identifiable molecular targets or biomarkers has hindered the development of treatment options in TNBC. Conventional chemotherapy, which primarily consists of anthracyclines and taxanes, is chosen as the mainstay systemic treatment for TNBC patients, despite resistance and relapse being the main problems associated with a poor prognosis [19]. Thus, it is crucial to investigate alternative biomarkers contributing to TNBC disease progression and prognosis, as they help establish effective individual therapy. Earlier, we identified BRIC5 (Baculoviral IAP Repeat Containing 5) as one of the many genes and biological systems associated with the numerous drug resistance mechanisms in TNBC patients [20]. Using natural compounds as an alternative treatment is safe, economical, and effective in managing various diseases, including cancer. Thymoquinone (TQ), obtained from the natural compound *Nigella sativa*, has emerged as the most promising compound for preventing and treating several conditions, including cancer [21]. TQ has been investigated for its antioxidant, anticancer, and anti-inflammatory activities in vitro and in vivo. Numerous studies have recently shown that TQ has anticancer properties against several cancer types, including breast, lung, laryngeal, ovarian, osteosarcoma, and myeloblastic leukemia. Among the significant anticancer mechanisms of TQ, according to a thorough review of the literature, are cell cycle arrest, inducing apoptosis by activating different caspases, inhibition of protein expression of antiapoptotic genes such as *Bcl-2*, *Bcl-Xl XIAP*, lower *AKT* phosphorylation, inducing p38 phosphorylation, and interfering with the survival of cells involving the PI3-K/Akt pathway [4]. Recently, we reported the anticancer mechanisms of TQ on MDA-MB-231 and MDA-MB-468 cell lines, including cell cycle arrest, apoptosis induction and regulation of several apoptosis-regulated genes, antiproliferation, anticlonogenicity, and antimetastasis [22].

The current study was designed to investigate the mechanisms by which the natural compound TQ affects cell viability, proliferation, and metastasis and modulates the release of proinflammatory cytokines in TNF-α-stimulated TNBC cells.

## 2. Results

### 2.1. Thymoquinone Decreases the Viability of TNF-α-Treated TNBC Cells

MDA-MB-231 and MDA-MB-468 cells were examined for viability in the presence of TQ to investigate TQ’s cytotoxic effects in TNF-α-stimulated cells (Figure 1). After being exposed for 24 h, both cell lines responded to the compound significantly (*p* < 0.05), starting at 5.0 µM for MDA-MB-231 cells and 10 µM for MDA-MB-468 TNBC cells, with nonsignificant effects at 5 µM for MDA-MB-468 cells. The viability was decreased by 14, 24, and 42% at 5, 10, and 15 µM, respectively, for MDA-MB-231cells. In contrast, at the same concentration for MDA-MB-468 cells, the viability decreased by 6, 10, and 28%, respectively. However, after 20 µm, the viability significantly decreased for MDA-MB-468 compared with MDA-MB-231, on which at 25, 30, 40, and 50 µm, the viability decreased by 82, 90, 117, and 118%, while for MDA-MB-231, it decreased by 64, 71, 75, and 83%, respectively, at the same concentrations, which indicates that increasing the concentrations of TQ shows more cytotoxic effect on TNF-α-stimulated MDA-MB-468 cells than MDA-MB-231 cells. The dose–response data analysis’s IC50 values, which were 26.86 ± 1.5 µM for MDA-MB-231 cells and 27.49 ± 1.4 µM for MDA-MB-468 cells, respectively, showed relatively similar inhibitory responses for TQ in these two models.

### 2.2. TQ Reduced TNF-α-Induced Proliferation of TNBC

The growth-inhibitory potency reported with longer exposure times served as evidence for the long-term effect of TQ on TNF-α-stimulated MDA-MB-231 and MDA-MB-468 TNBC cells in antiproliferative tests. The findings indicated that the proliferation rate was lowered in a time- and concentration-dependent manner. After 48, 72, and 96 h of treatment, TQ dramatically reduced cell proliferation in both TNF-α-stimulated cell lines relative to each exposure period. At a concentration of 5 µM, the proliferative rate in the MDA-MB-231 cell (Figure 2A) decreased by 14, 46, and 22% after 48, 72, and 96 h of exposure, respectively. At the same concentration, the growth rate was reduced in its counterpart TNF-α-stimulated MDA-MB-468 cell (Figure 2B) by 3, 15, and 34%, respectively. Compared with MDA-MB-468 cells, the antiproliferative effect of TQ on MDA-MB-231 cells exhibits a similar response pattern starting at 20 µM regardless of exposure time or concentration. Overall, the TNF-α-stimulated MDA-MB-231 cell line response to the antiproliferative effect of TQ was greater than that of MDA-MB-468 cells, consistent with the findings of the viability investigation. These effects’ variability may indicate the different anti-TNBC mechanisms of TQ in these two genetically distinct cell lines.

### 2.3. TQ Inhibited TNF-α-Induced Clonogenicity of TNBC

Next, we investigated the potential of TQ to affect the clonogenicity ability of TNBC cells. The survival of clonogenic cells was significantly inhibited when the TNF-α (50 ng/mL)-stimulated cells were exposed to TQ (0–50 µM) at 1, 3, and 6 h. TQ had a significant time- and concentration-dependent effect on the number of colonies, which was evidence of its effects on TNF-α-stimulated TNBC cell colony formation. At 5 µM, the colony formation of TNF-α-stimulated MDA-MB-231 TNBC cells are reduced significantly by 12, 13%, and 10% and by 32, 32, and 53% at 10 µM for the exposure periods of 1, 3, and 6 h (*p* < 0.05), respectively. From 15 µM onwards, there is a comparable concentration-time-dependent decrease in the colony formation ability of the stimulated MDA-MB-231 TNBC cells (Figure 3A). In contrast, in TNF-α-stimulated MDA-MB-468 TNBC cells at 5 µM of TQ, the colony formation was reduced by 80%, and at 10 µM and onwards, irrespective of time exposure and concentration, the colony formation was reduced by more than 90% (*p* < 0.0001) (Figure 3B). The findings further confirmed that TNF-α-triggered TNBC cells respond similarly to TQ treatment. TNF-α-stimulated MDA-MB-231 TNBC cells responded to TQ less sensitively against clonogenicity than TNF-α-stimulated MDA-MB-468 TNBC cells.

### 2.4. TQ Inhibits TNF-α-Induced Migration of TNBC Cells

Chemokines are crucial in tumor progression at distant sites by promoting tumor migration and metastasis [3]. A migration experiment was conducted to identify the possible antimigratory impact of TQ in TNF-α-stimulated TNBC cells (Figure 4). We chose TQ concentrations for this experiment, allowing for ≥80% survivability (Figure 1). In both cells under observation, a proportionate link between the gap width and exposure time was found, as seen in the figures. We incubate both cell lines until the stimulated cells are completely sealed. The wound was completely healed in the stimulated wells after 48 h for MDA-MB-231 cell lines (Figure 4A) and 72 h for MDA-MB-468 cell lines (Figure 4D). As a result, a quantifiable significant inverse association between the cotreated and the stimulated percentage of migratory cells was found. By comparison, after 24 and 48 h at 10 µM TQ, the migrated TNF-α-stimulated MDA-MB-231 cells showed a 6.7- and 2.3-fold change inhibition (*p* < 0.001), respectively (Figure 4B,C). At 10 µM TQ, their counterpart MDA-MB-468 cells demonstrated inhibition of 1.93-, 2.5-, and 1.9-fold changes after 24, 48, and 72 h, respectively (*p* < 0.001) (Figure 4E–G). According to the data, TQ may have more potent antimetastatic effects in TNF-α-stimulated MDA-MB-231 cells than in MDA-MB-468 cells.

### 2.5. TQ Inhibits the Release of Chemokines in TNBC Cells

TQ’s antichemokine effect on inflammation was determined in TNF-α-stimulated TNBC cells using the human cytokine membrane array. On the captured blots, the levels of cytokines released by the treated groups were represented as dots of varying intensities (Figure 5). The highest spot intensities were visible in blots containing TNF-α-stimulated cell supernatant. Compared with other chemokines, CCL3 and CCL4 in MDA-MB-231 cells (Figure 5B) and CCL2 and CCL20 in MDA-MB-468 cells (Figure 5H) were noticeably altered in their expression (highlighted with red and blue line frames). Spot intensities were converted to numbers utilizing cytokine data analysis software (Ray Biotech, Norcross, GA, USA), which supports the spot data obtained (Figure 5E,F,K,L). The semiquantitative standardized data demonstrated that both TNF-α-stimulated TNBC cell lines significantly increased CCL2, CCL3, CCL4, and CCL20 levels (*p* < 0.001). Moreover, TNF-α significantly increased CCL3 and CCL4 activation in MDA-MB-231 cells by a 3.6- and 2.4-fold change, while TQ significantly reduced the stimulated cells by 2.1- and 1.7-fold (*p* < 0.001), respectively. In contrast, TNF-α increased the expression of CCL2 and CCL20 in MDA-MB-468 cells by a 28.1- and 9.9-fold change (*p* < 0.001), and TQ significantly reduced the expression of CCL2 and CCL20 in the presence of TNF-α by a 15.6- and 3.3-fold change (*p* < 0.01) in MDA-MB-468 cells, respectively. Interestingly, only TQ-treated MDA-MB-231 cells showed a modest but significant suppression of CCL3 by 0.5-fold change (Figure 5F). In contrast, in CCL2 and CCL20 in MDA-MB-468 and CCL3 in MDA-MB-231 cells, TQ-only treatment showed no significant difference compared with the control (Figure 5K).

### 2.6. TQ Inhibited Induction of Chemokines in TNF-α-Stimulated TNBC Cells

CCL2, CCL3, CCL4, and CCL20 protein expression in the supernatant of each sample was quantified and validated using ELISA experiments (Figure 6A,B). The obtained ELISA data for both cell lines confirmed the blot analysis results. MIP-1α (CCL3), which was highly elevated in TNF-α-stimulated MDA-MB-231 cells (*p* < 0.0001), was decreased by 1.7-fold (*p* <0.01). MIP-1β (CCL4) was dramatically downregulated by TQ (10 µM) when cotreated in TNF-α-stimulated MDA-MB-231 cells by 1.2-fold. When TNF-α-stimulated MDA-MB-468 cells were cotreated with 15 µM TQ, the elevated MCP-1 (CCL2) was considerably reduced by 4.5-fold. When TQ was added with TNF-α-stimulated MDA-MB-468 cells, the protein expression of MIP-3α (CCL20) was decreased by 1.5-fold. Interestingly, same as the blot result, only TQ-treated MDA-MB-231 cells showed a modest but significant suppression of CCL3 by 1-fold change (Figure 5A). In contrast, TQ-only treatment showed no significant effect in CCL2 and CCL20 in MDA-MB-468 and CCL3 in MDA-MB-231 cells.

### 2.7. TQ Represses the mRNA Expression of the Chemokines in TNF-α-Stimulated TNBC Cells

TQ’s effect on releasing the chemokines CCL2, CCL3, CCL4, and CCL20 in TNBC cells was evaluated using RT-qPCR. The findings from the data showed that the expressions of all four genes’ mRNAs were consistent with those of cytokine microarray and ELISA protein analysis (Figure 7). Each TNBC cell type under investigation increased its response to TNF-α ± TQ. The mRNAs for CCL3 and CCL4 were significantly upregulated by TNF-α (*p* < 0.0001, Figure 7A,B). TQ decreased the expression of these genes by 1.3 and 1.8-folds in MDA-MB-231 stimulated cells, respectively (*p* < 0.01, Figure 7A,B). In contrast to the cytokine membrane array and ELISA result, TQ did not demonstrate any significant effect on CCL3 in MDA-MB-231 TNBC cells that were not stimulated. Nevertheless, TQ resulted in a 1.5-fold change (*p* < 0.01, Figure 7C) in suppression of the CCL2 gene in TNF-α-treated MDA-MB-468 cells, along with a 6.7-fold decrease in CCL20 mRNA (*p* < 0.0001, Figure 7D). Additionally, only TQ-treated TNBC cell models demonstrated a significant reduction in the CCL2 (*p* < 0.05) and CCL20 (*p* < 0.001) genes by 1.5- and 6.1-fold inhibition. In general, the obtained PCR data supported our cytokine array and ELISA data, which showed a consistent expression in TNF-α-stimulated TNBC cells.

### 2.8. TQ Represses the Protein Expression of the Chemokines in TNF-α-Stimulated TNBC Cells

Using capillary electrophoresis Western analysis, we further examined the protein expression of CCL2, CCL3, CCL4, and CCL20 in TNF-α-stimulated TNBC cells (Figure 8). In the sample lysates of the four groups—control, TQ-treated, and TNF-α + TQ treated cells—the expressions of the four concerned proteins were assessed. TNF-α-treated TNBC cells displayed significantly elevated levels of CCL3, CCL4, CCL2, and CCL20, as shown in the Western bands (Figure 8A,C,E,G, respectively) and Compass software data analyses (Figure 8B,D,F,H). The cotreated samples of TQ then suppressed these four elevated proteins (*p* < 0.05). TNF-α and TQ-cotreated cells significantly reduced CCL3 and CCL4 by 1.9-fold in MDA-MB-231 (*p* < 0.01, Figure 8B,D). In contrast, the CCL2 and CCL20 levels were significantly inhibited in the TNF-α + TQ-treated MDA-MB-468 cells by a 12.5- and 1.3-fold change suppression, respectively (Figure 8F,H). Interestingly, TQ treatment only on MDA-MB-231 and MDA-MB-468 TNBC cells showed no noticeable differences from the nonstimulated control groups. These could explain our earlier findings and point to TQ’s potential antichemokine activities associated with inflammation.

## 3. Discussion

TNBC, which makes up nearly 20% of breast cancers, has no HER2/neu receptor, ER, or PR expression. TNBC treatment is challenging due to the limited therapeutic available options, poor prognosis, aggressive tumor behavior, and absence of targeted therapy [23,24]. TNBC is also significantly more invasive and metastatic than breast cancers that are receptor-specific [24]. Thus, finding targeted anticancer agents and promising treatment biomarkers is crucial. Researchers have hypothesized that using natural compounds can reduce the generation of oncogenic cytokines while also benefiting treatment for aggressive, inflammatory breast tumors [25,26]. TNBC and other types of tumor cells have shown anticancer activity in response to the compound TQ [4,22]. Both in vitro and in vivo settings have been used to study TQ’s anticancer properties. Among the significant anticancer mechanisms of TQ, according to our review of the literature, are cell cycle arrest, inducing apoptosis by activating different caspases, including caspase-8 and 9, inhibiting protein expression of antiapoptotic genes such as Bcl-2, Bcl-Xl XIAP, surviving lower AKT phosphorylation, inducing p38 phosphorylation, and interfering with the survival of cells involving the PI3-K/Akt pathway [4]. The anticancer effects of TQ on the MDA-MB-231 and MDA-MB-468 cell lines have recently been reported [22]. We have also reported on TQ effects which include cell cycle arrest, induction of apoptosis, regulation of numerous apoptosis-regulated genes, inhibition of proliferation, antagonism of clonogenicity, and inhibition of metastasis [22].

Pathological inflammatory cytokines and inflammation are significantly associated with TNBC [27]. TNF-α is a cytokine that is frequently overexpressed in inflammatory disorders. To cause inflammation and promote cellular proliferation, TNF-α can activate several signaling pathways. Upregulated TNF-α in human TNBC stimulated the growth of the mammary molecules and activated many genes involved in the cancer cells’ growth, invasion, and metastasis [28]. More critically, elevated TNF-α production may also result in the release of a variety of critical chemokines that play a crucial role in tumorigeneses [29].

The cytotoxic study findings showed that TNF-α stimulated the growth of MDA-MB-468 compared with MDA-MB-231 cells when using TQ at the lowest and modest concentrations; MDA-M-231 cells showed more sensitivity to TQ compared with MDA-MB-468. The reverse is true when increasing the concentrations of TQ for MDA-MB-468 TNBC cells. This finding confirms that TNF-α-stimulated MDA-MB-468 cells are more proliferative than MDA-MB-231. The proliferation of TNF-α-stimulated MDA-MB-231 and MDA-MB-468 TNBC cells was dramatically reduced by TQ in the current study. According to previous studies, chemokines also modulate stromal and tumor cells, controlling cancer cell survival, proliferation, and angiogenesis. In particular, CCL2 and CCL3 have been found to have direct protumor effects by promoting tumor invasion and proliferation, survival, motility, and stemness and preventing apoptosis [30,31]. TNBC can develop from the inability of proapoptotic proteins to function properly or from the overabundance of antiapoptotic proteins [32]. Considering what we have discovered, TQ may effectively reduce protumor chemokines and inhibit inflammation-driven TNBC cell proliferation.

We explored the colony formation assay in a way analogous to further study of the inhibitory effect of TQ on the tumor growth of TNF-α-activated TNBC cells. Our results verify that TQ considerably reduced the colony formation of TNBC cells and further demonstrated its potential to inhibit TNBC proliferation.

We further evaluated the compound’s capacity to halt migration in TNF-α-stimulated TNBC cells. Our migration analysis suggested the antimigratory activity of TQ in both cell lines under investigation with relatively higher effects in MDA-MB-231 cells (Figure 4). Chemokines are cohesively linked to metastasis, fostering tumor progression at distant sites [33]. Migration is the key to cancer-related death [34] since, in the migration process, tumor cells segregate from the initial tumor and invade surrounding or distant tissues to establish a secondary tumor [35]. More than one-third of patients with TNBC are at risk for metastasis during their disease [36]. Furthermore, 46% of TNBC patients will have a distant metastasis [24]. Unfortunately, patients with metastatic TNBC have a poor prognosis, with 3.3 months of treatment [37]. Therefore, it is essential to consider our finding that the natural compound TQ can inhibit metastasis.

In this study, TNF-α stimulation increased the expression of the chemokines in both TNBC cell lines (Figure 4). Chemotactic chemokines, MCP-1/CCL2, and macrophage inflammatory protein-1α, -1β (MIP-1α/CCL3 and MIP-1β/CCL4), and MIP-3α/CCL20 are the most common chemokines significantly altered by the natural compound TQ. In our study, CCL2, CCL3, CCL4, and CCL20 mRNA and protein expressions in TNF-α-activated MDA-MB-231 and MDA-MB-468 cells were drastically downregulated in the presence of low concentrations of TQ (10 and 15 µM). Many studies have shown that chemokines and their receptors play a role in the development and spread of tumor cells and their proliferation and survival by altering the ratio of pro- and antiapoptotic proteins in tumor cells [38,39,40,41].

The chemokine CCL2/MCP-1 (macrophage chemotactic protein-1) is highly expressed in breast tumors and stromal cells. CCL2 is highly expressed by tumor-associated macrophages (TAM) in the tumor stroma, and substantial TAM accumulation is associated with disease recurrence and a poor prognosis [42]. Neutralizing antibodies to CCL2 further inhibited the development of lung metastases in mice receiving a xenograft of the CCL2-expressing MDA-231 human breast cancer [43]. Research implies that CCL2 helps breast cancer cells spread through the body [44]. The cancer-associated membrane glycoprotein dysadherin facilitates the invasion of tumor cells into the Matrigel in estrogen receptor (ER)-negative breast tumor cells through controlling CCL2 production in vitro and lung metastases in an in vivo animal model. Hence, dysadherin overexpression may promote the growth of breast cancer by increasing CCL2 expression [45]. By enhancing the expression of CCL2, mesenchymal stem cells promote cell proliferation in heterogeneous TNBC [46]. According to previous similar studies, BC lung metastasis can be promoted by inflammatory monocyte recruitment through CCL2 expression by metastatic tumor cells and the lung stroma [47].

CCL2 and CCL3 can promote the invasion and spread of tumors. CCL2 influences tumor vascularization and metastasis by targeting vascular endothelial cells through the Janus kinase 2 (JAK2)-STAT5 and p38 mitogen-activated protein kinase pathways [48,49]. By causing monocytes to secrete the matrix metalloproteinase 9 (MMP9) enzyme, CCL2 and CCL3 can promote the extravasation of tumor cells [50]. Moreover, CCL2 can support cancer cell motility, proliferation, survival, and the epithelial–mesenchymal transition (EMT) [51,52]. 

Chemokine (C-C motif) ligand 4, also known as CCL4 or macrophage inflammatory protein-1 (MIP-1), is essential for the control of the immune response, inflammation, and the development of cancer [53]. Data suggest that tumor-cell-derived CCL4 promotes BC metastasis by prompting CCR5-expressing fibroblasts to express connective tissue growth factor [54]. CCL4 polymorphisms may increase susceptibility to oral cancer, as serum levels of CCL4 have been observed to be considerably more significant in patients with head and neck squamous cell carcinoma than in controls [55,56].

Macrophage proinflammatory chemokine-3α (MIP-3/CCL20) is overexpressed in pancreatic carcinoma cells and infiltrates macrophages near tumors. MIP-3α/CCL20/promotes TAM migration, stimulating cancer cell proliferation [57]. Compared with healthy individuals, CCL20 levels were higher in BC, attracting immune cells that express CCR6 near the tumor [58]. The CCL20 that tumor cells release encourages the migration and growth of neighboring breast cells and the initiation of BC [59]. According to studies, CCL20 is involved in BC’s lung and bone metastases, and higher levels of CCL20 expression are associated with poorer overall survival [60,61]. Moreover, CCL20 expression was higher in TNBC than non-TNBC status and AA with BC than in European American [62]. Chemotherapy has been demonstrated to increase CCL20 expression, which promotes chemoresistance and further suggests that CCL20 may play a role in therapeutic outcome discrepancies [63]. TQ dramatically reduced the mRNA and protein expressions in the current study and may offer exciting alternatives for treating and preventing TNBC, either alone or in combination with chemotherapy.

## 4. Materials and Methods

### 4.1. Materials and Reagents

Alamar Blue^®^ (a resazurin fluorescent dye solution) and TQ (purity ≥ 98%, cat# 27466-5G, Lot# MKCC0600) were purchased from Sigma-Aldrich (St. Louis, MO, USA). Trypsin-EDTA solution, penicillin/streptomycin, and phosphate-buffered saline (PBS) were purchased from the American Type Culture Collection (ATCC; Manassas, VA, USA). In this experiment, we used a DNA-freeTM kit from Life Technologies, Inc. (cat. No. AM1907, Thermo Fisher Scientific, Inc., Waltham, MA, USA). Fetal bovine serum (FBS) and Dulbecco’s modified Eagle’s medium (DMEM) were procured from VWR International (Radnor, PA, USA). We obtained cell culture flasks from Santa Cruz Biotechnology, Inc. (Dallas, TX, USA) and Thermo America Scientific (Ocala, FL, USA) cell culture plates. Moreover, we obtained Tumor Necrosis Factor-Alpha, Human Recombinant; 50 µg (cat# 228-11521-2), Human Cytokine Array C1000 (4); 4 Sample Kit (cat# AAH-CYT-1000-4), MIP-3alpha (Cat# ELH-MIP-3a, lot # 315230165), MIP-1beta (ELH-MIP1b, Lot # 0315230164), MIP-1alpha (Cat# ELH-MIP-1a, Lot # 0315230286), and MCP-1 (Cat# ELH-MCP-1, Lot# 0315230162) from RayBiotech (Peachtree Corners, GA, USA). SsoAd-vancedTM Universal SYBR^®^ Green Supermix, and iScriptTM cDNA Synthesis kit (cat. No. 170-8890), PrimePCR™ SYBR^®^ Green Assay: CCL2, Human desalted 200 × 20 µL reactions (UniqueAssayID: qHsaCID001160 8), PrimePCR™ SYBR^®^ Green Assay: CCL20, Human desalted 200 × 20 µL reactions (UniqueAssayID: qHsaCID001177 3), PrimePCR™ SYBR^®^ Green Assay: CCL4, Human desalted 200 × 20 µL reactions (UniqueAssayID: qHsaCED00442 60), PrimePCR™ SYBR^®^ Green Assay: CCL3, Human desalted 200 × 20 µL reactions (UniqueAssayID: qHsaCED00386 33), and PrimePCR™ SYBR^®^ Green Assay: GAPDHS, Human desalted 200 × 20 µL reactions (UniqueAssayID: qHsaCID001546 4) were purchased from Bio-Rad (Hercules, CA, USA). Goat antirabbit IgG H&L (HRP) (cat# ab6721), rabbit polyclonal to macrophage inflammatory protein 3 alpha (cat# ab9829), rabbit monoclonal (EP521Y) to CCL4/MIP-1 beta (cat# ab45690), rabbit monoclonal (EPR16618-90) to MIP-1 alpha/CCL3 (cat# ab179638), HRP rabbit monoclonal (EPR16891) to GAPDH—loading control (cat# ab201822), and rabbit monoclonal (EPR21025) to MCP-1 (cat# ab214819) were purchased from Abcam (Boston, MA, USA).

### 4.2. Cell Culture

MDA-MB-231 and MDA-MB-468 were bought from ATCC and handled per the company’s instructions. Both cell lines were grown in 75 mL tissue culture flasks as monolayers at 37 °C in a humidified 5% CO_2_ incubator, subculturing as necessary with trypsin/EDTA (0.25%). In total, 4 mM L-glutamine, 10% heat-inactivated FBS (*v*/*v*), and 1% penicillin/streptomycin salt solution (100U/mL and 0.1 mg/mL, respectively) was added to the complete growth DMEM. Then, 2.5% heat-inactivated FBS was added to DMEM, the experimental medium, for the study [20].

### 4.3. Cytotoxicity Assay Using Alamar Blue

Alamar Blue^®^ (Sigma-Aldrich, St. Louis, MO, USA) was used to test MDA-MB-231 and MDA-MB-468 cells for TQ’s cytotoxicity on TNBC cells [22]. First, 5 × 10^5^ cells/100 μL/well of cells were plated in 96-well plates and then incubated at 37 °C overnight. Fifty ng TNF-α ± TQ was dissolved in DMSO and applied to both cell lines for 24 h at concentrations ranging from 0 to 50 μM. Wells that had undergone the same treatment but lacked TQ were utilized as “blanks”. Three duplicates of each experiment were run. Following a predetermined exposure time, 20 μL of Alamar Blue^®^ at a concentration of 0.5 mg/mL were added to each well, and they were then incubated for 4 h at 37 °C. Resazurin, the active component of the Alamar Blue reagent, was measured using a Synergy HTX Multi-Mode microplate reader at an excitation/emission wavelength of 530/590 nm (BioTek Instruments, Inc., Winooski, VT, USA). Cell viability and fluorescence signal intensity are correlated. FDA-recommended standards for examining various biological effects were followed while choosing TQ concentrations that limit cell viability by about 30% [22].

### 4.4. Cell Proliferation Assay

The Alamar Blue^®^ assay evaluated TQ’s effect on cell proliferation in TNF-α-stimulated MDA-MB-231 and MDA-MB-468 TNBC cells [20]. On 96-well plates, cells were seeded at a density of 1 × 10^4^ cells/100 μL/well and incubated at 37 °C overnight. A total of 50 ng TNF-α ± TQ was applied to the MDA-MB-231 and MDA-MB-468 cell lines in a final volume of 200 μL/well for 48, 72, and 96 h at concentration ranges of cytotoxicity assay (0–50 μM). Equivalent wells devoid of cells were used as a blank. After each exposure period, 20 L of Alamar Blue^®^ was added to each well. The plates were then incubated for 4 h at 37 °C before being read with a Synergy HTX Multi-Mode microplate reader (BioTek, Inc., Winooski, VT, USA).

### 4.5. Colony Forming Assay

The previously described method with modification was used to conduct a clonogenic experiment to evaluate the impact of TQ on TNF-α-stimulated MDA-MB-231 and MDA-MB-468 TNBC cells [20]. On 96-well plates, both cell lines were seeded at a density of 1 × 10^4^ cells/100 mL/well and handled in the same manner as the previous proliferation study. Following the exposure time (3 or 6 h), the DMSO/ TNF-α ± TQ-containing media were aspirated, and all wells were given two PBS rinses. The colonies formed in both treated and untreated cells were then evaluated using the Alamar Blue^®^ reduction assay, which was previously covered in the cell viability assay after the cells had grown for seven days. Each assay was run in triplicates.

### 4.6. Migration Assay

Migration assay was run using two-well self-insertion kits (Ibidi^®^ and VWR) using previously described procedures [22]. The 2-well silicone insert was placed into each well of the 6-well plate. Overnight, 3.5 × 10^4^ MDA-MB-231 and MDA-MB-468 cells were sown in a 70 µL medium compartment. The inserts were gently taken out, the media containing the floating cells were decanted, and all the wells were given two PBS washes. After that, cells were given doses of 50 ng TNF-α ± TQ (10 μM for both cells). The plates were put in the cell culture incubator until the space between the TNF-α-stimulated wells was completely sealed. The results were recorded after a 24, 48, and 72 h exposure period, and each well’s gap width was measured. The test was run in triplicates with n = 3 for each treatment group. The following equation was used to calculate the percentage of moved cells.
$$\left(1-\frac{A2}{A1}\right)*100$$

The difference in migration of vehicle control cells and treated cells were used to quantify TQ’s prevention of cancer cell migration, i.e., migration inhibition = $\left(\frac{A2\left(\mathrm{treatment}\right)}{A1}\right)-\left(\frac{A2\left(\mathrm{control}\right)}{A1}\right)$.

### 4.7. Human Cytokine Antibody Arrays Membrane

MDA-MB-231 and MDA-MB-468 TNBC-treated cells were used to analyze the expression of cytokine-mediated proinflammatory molecules using a human cytokine antibody array (Ray Biotech, AAH-CYT-1000, Norcross, GA, USA). We used a set of four 75 cm^2^ TC flasks that were seeded with 10 106 cells each for each cell line. The flasks were divided into four groups: control, TQ-treated, TNF-α-stimulated, and cotreated (TNF-α + TQ). The concentrations for TQ were selected based on the cytotoxicity assay results with a negligible effect on cell survival. As a result, TQ-treated cells were exposed to either 10 µM or 15 µM in MDA-MB-231 and MDA-MB-468 cells, respectively, while control cells only received the greatest concentration of DMSO (0.1%). In the third flask set, 50 ng/mL of TNF-α was evenly administered to each cell line. The cotreated cells were next subjected to a cocktail of 50 ng/mL TNF-α and TQ at 10 µM or 15 µM for MDA-MB-468 cells and MDA-MB-231 cells, respectively. Cells were mechanically collected in a new set of falcon tubes and centrifuged after 24 h of exposure. The appropriate cell pellet and cell-free supernatant from each sample were kept at 80 °C for future research. The expression of several cytokines that mediate proinflammatory molecules was assessed using a semiquantitative test using antibody-coated array blots in the previously gathered cell-free supernatant. The four membranes for each set were initially carefully placed in the designated tray without touching the surface that was coated with the antibody. An amount of 2 mL of the supplied buffer was used to block each membrane equally. Following the aspiration of the blocking solution, 1 mL of recently defrosted cell-free supernatant was pipetted onto the appropriate blot. The tray was shaken while being maintained overnight at 4 °C. After extracting supernatants from each chamber, the membranes were washed with the kit buffers. All membranes were then exposed to 1 mL of freshly diluted biotinylated antibody for 2 h at room temperature (RT), followed by a second wash. Following a final wash, the membranes were recently exposed for a further 2 h to freshly made horseradish peroxidase-conjugated streptavidin (HRP-Streptavidin). The cytokine intensities on the different blots were determined using a designated chemiluminescence buffer. Using a Flour-S Max Multi-imager, the blot images were displayed in less than 5 min (Bio-Rad Laboratories, Hercules, CA, USA). The Quantity One Program was used to quantify the spot intensities (Bio-Rad Laboratories, Hercules, CA, USA). The cytokine intensities for the Human Cytokine Array AAH-CYT-1000 were normalized using Excel-based data analysis software (Microsoft Excel 97-2003 worksheet).

### 4.8. Human Chemokine ELISA and Quantification

MDA-MB-231 and MDA-MB-468 TNBC cells were subjected to various treatments for 24 h and had their supernatants collected. The following treatments were used: control (cells only), TQ (75 μM), TNF-α (50 ng/mL), and TQ (10 μM) + TNF-α (50 ng/mL). The Raybiotech ELISA CCL2, CCL3, CCL4, and CCL20 kit instructions were used for the experiment. TNBC cells were obtained and centrifuged for 4 min at 4 degrees Celsius at 1000 rpm. According to the manufacturer’s recommendations, a particular ELISA for the detection of human CCL2, CCL3, CCL4, and CCL20 was used. In brief, 100 μL of the supernatants from each standard and sample were put into 96-well plates that were precoated with capture antibody, and they were then incubated for 2.5 h at room temperature while being shaken. After washing, a 100 μL combination of biotinylated antibodies was made, poured into each well, and allowed to sit for one hour. The mixture was then aspirated, and 100 μL of streptavidin solution was added to each well before it was incubated for 45 min. After adding 50 μL of stop solution, 100 μL of substrate reagent was pipetted into each well and incubated for 30 min. The optical density of the samples was determined at 450 nm using a Synergy HTX Multi-Reader (BioTek, Winooski, VT, USA).

### 4.9. Gene Expression Study

#### 4.9.1. RNA Extraction

Profiling the mRNA expression of genes was developed in MDA-MB-231 and MDA-MB-468 TNBC cells using previously described procedures [40]. In brief, each cell line was grown overnight at 37 °C (at a density of 10 × 10^6^ cells/mL in T-75 flasks) before TNF-α ± TQ was applied for 24 h. After a 24 h treatment, cells were mechanically removed from each flask, pelleted, and given two PBS washes. According to the manufacturer’s instructions, each sample was first homogenized with 1 mL of Trizol^®^ (Thermo Fisher Scientific, Inc., Waltham, MA, USA) before 200 µL of chloroform was added (Sigma Aldrich, St. Louis, MO, USA) to extract total RNA. Following a 15 min vortexing period, samples were centrifuged for 15 min at 10,000× *g* and 8 °C. The top RNA-rich layer was transferred to a clean tube and treated with 500 µL of 2-propanol to yield RNA pellets. The liquid portion of each tube was aspirated after 20 min of centrifugation at 10,000× *g*, and the pellets were then washed in 75% ethanol, dried by air, and reconstituted in water devoid of nuclease.

#### 4.9.2. Complementary DNA (cDNA) Synthesis

We used a nanodrop spectrophotometer to measure the concentration of reconstituted RNA in each sample before beginning the cDNA synthesis process. The DNA-freeTM kit from Thermo Fisher Scientific (Thermo Fisher Scientific, Inc., Waltham, MA, USA) was used, and it involved incubating RNA at a concentration of 200 µg/mL with a 1× DNase cocktail for 30 min at 37 °C. A DNase inactivator was added to stop the process. The samples were centrifuged for 3 min at 9000 rpm to collect the DNA-free supernatant. We used the iScriptTM cDNA Synthesis kit and the instructions supplied to produce cDNA (Bio-Rad Laboratories, Hercules, CA, USA). In each well of the 96-well PCR plates, 9 µL of nuclease-free water, 5 µL of DNA-free supernatant, and 6 µL of an advanced reaction mix reverse transcriptase (RT) cocktail were added, for a total volume of 20 µL. Using the CFX96 Touch Real-Time PCR Detection System, the RT reaction was started at 46 °C for 20 min, then inactivated at 95 °C for 1 min (Bio-Rad). The obtained cDNA plates were immediately placed in a freezer at −80 °C.

#### 4.9.3. Quantitative Reverse Transcription–Polymerase Chain Reaction (qRT-PCR)

Using the CFX9 Real-Time System (Bio-Rad Laboratories, Inc. Hercules, CA, USA), the CCL2, CCL3, CCL4, and CCL20 expressions were assessed. At a final volume of 20 µL, 50 ng/sample of the previously synthesized cDNA was mixed with two real-time Master Mix solutions. A 95 °C incubation period of 2 min was followed by 39 amplification cycles to begin the PCR cycle. Each cycle includes 10 s of denaturation at 95 °C, 30 s of annealing at 60 °C, and 5 s of a melting curve at 65–95 °C. Using GAPDH as a reference gene, the mRNA regulation for the examined genes was normalized. For each primer, three experiments were used to generate the evaluated data.

### 4.10. Capillary Electrophoresis Western Analysis 

Protein expression was measured using Wes fully Automated Western System ver.6.0 (Protein Simple, San Jose, CA, USA). The design of the cell setup and treatment was identical to 4.9.1. Each sample’s cell pellet was lysed using a protease inhibitor and lysis buffer cocktail. The protein concentration in cell lysates was determined using the PierceTM BCA protein assay kit’s instructions (LOT# 49685—2-40kDa, LOT# 95272-12-230kDa, Thermoscientific, Rockford, IL, USA). CCL2. CCL3, CCL4, CCL20, GAPDH primary, and goat antirabbit secondary antibodies were used for band immunodetection. The endogenous control antibody was rabbit mAb, while the secondary antibody was goat antirabbit IgG, HRP-linked antibody. Abcam supplied the primary and secondary antibodies (Beverly, MA, USA). The protein expression in each sample was assessed using a Wes Separation Module 25 capillary cartridge and its matching Anti-Rabbit Detection Module kit. Optimal concentrations for the cell lysates and antibodies under examination were established following Protein Simple’s procedure. The applied concentration of the cell lysates for both cell types was 1 mg/mL. The dilution factor for the 4 antibodies under examination was 1:100. We used the endogenous control GAPDH to normalize the data. The results were obtained from three independent experiments, and Prism-GraphPad ver. 9.0. (San Diego, CA, USA) was used for analysis.

### 4.11. Statistical Analysis

The present study’s data are expressed as the means ± standard error of the mean (SEM). Prism-GraphPad software analysis generated the IC_50_ values by nonlinear regression model of log (inhibitor) vs. normalized response variable slope with the R^2^ best fit and the lowest 95% confidence interval. An unpaired *t*-test was used to compare the two groups. Comparisons of more than two groups were determined using a one-way analysis of variance (ANOVA) followed by a Bonferroni post-test. In all studies, *p* < 0.05 indicates a statistical significance difference. As mentioned in each figure’s caption, a detailed data analysis was performed.

## 5. Conclusions

Our study showed that compared with TQ-non-treated TNF-α-stimulated cells, TQ dramatically reduced TNF-α-stimulated TNBC cells’ survival, proliferation, colony formation, and migration. Proinflammatory chemokine activation is strongly associated with tumor cell survival, growth, colony formation, and migration. The changed chemokines in TNF-α activated MDA-MB-231 and MDA-MB-468 TNBC cells are CCL2, CCL3, CCL4, and CCL20. In conclusion, the results of this study indicate that genetically different cell lines may respond to the natural compound TQ in various ways, as TQ targets CCL3 and CCL4 in MDA-MB-231 cells and CCL2 and CCL20 in MDA-MB-468 cells. Therefore, these results indicate that TQ may be recommended as a component of the therapeutic strategy for TNBC treatment. However, more studies are needed to investigate the antichemokine effects of TQ in TNBC in vivo models.

## Figures and Tables

**Figure 1 ijms-24-09878-f001:**
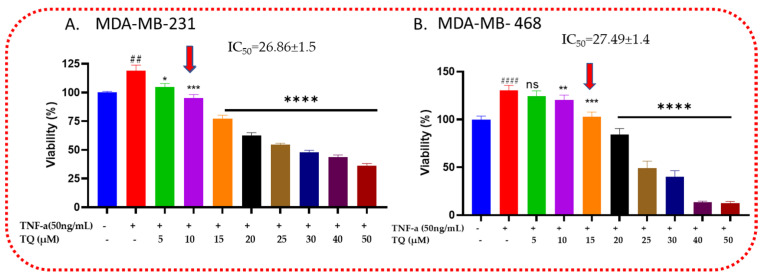
Triple-negative breast cancer (TNBC) cells stimulated by TNF-α are shown to be cytotoxic by TQ: TNF-α (50 ng/mL) and TQ (0–50 µM) were cotreated for 24 h on MDA-MB-231 and MDA-MB-468 cells. Each data point corresponds to the average and standard error of the mean of the three independent experiments with n = 6. Using GraphPad Prism, the percentages of cell survival were adjusted to the TNF-α-stimulated cells. The red arrows on the x-axis represent the specified concentrations for the next cytokine array study (10 µM for MDA-MB-231 and 15 µM for MDA-MB-468 TNBC cells). The statistical difference between the stimulated and cotreated groups was confirmed using a one-way ANOVA followed by the Bonferroni multiple comparison test. ns: nonsignificant, whereas # compared with control and * compared with TNF-α stimulated. * *p* < 0.05, ** *p* < 0.01, ## *p* < 0.01, *** *p* < 0.001, #### *p* < 0.0001, and **** *p* < 0.0001.

**Figure 2 ijms-24-09878-f002:**
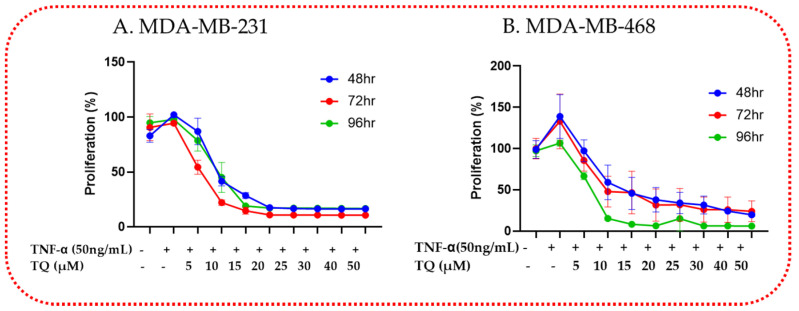
TQ reduces cell proliferation in TNBC cells using Alamar Blue: Both cell lines, MDA-MB-231 (**A**) and MDA-MB-468 (**B**), were plated in 96-well plates at 1 × 10^4^ cells/100 µL/well and treated similarly for 48, 72, and 96 h in a time-dependent manner with TNF-α (50 ng/mL) ± TQ at concentration ranges of 0–50 µM. Each data point presents the average ± SEM of three independent experiments, n = 6. One-way ANOVA was used to determine the significance of the difference between the control and treated groups.

**Figure 3 ijms-24-09878-f003:**
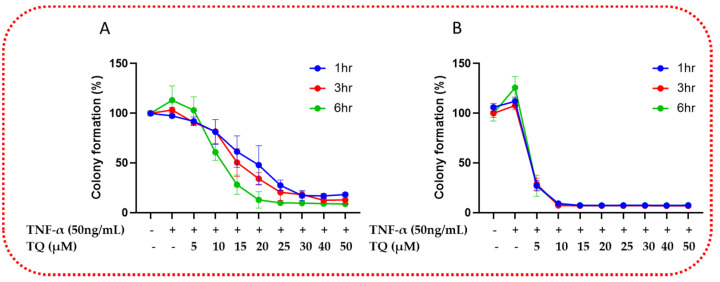
TQ treatment diminishes the clonogenic potential of TNF-α-stimulated TNBC cells: TQ’s colony formation effects over a prolonged period in MDA-MB-231 (**A**) and MDA-MB-468 (**B**) cells were shown by performing a colony formation assay. The TNF-α-stimulated TNBC cell lines were allowed to grow for seven days after treatment with 0–50 μM TQ for 1, 3, and 6 h. Graphs show the relative colony formation ability of TNF-α-stimulated TNBC cells cotreated with TQ compared with stimulated only. All error bars represent the standard error of the mean (n = 6).

**Figure 4 ijms-24-09878-f004:**
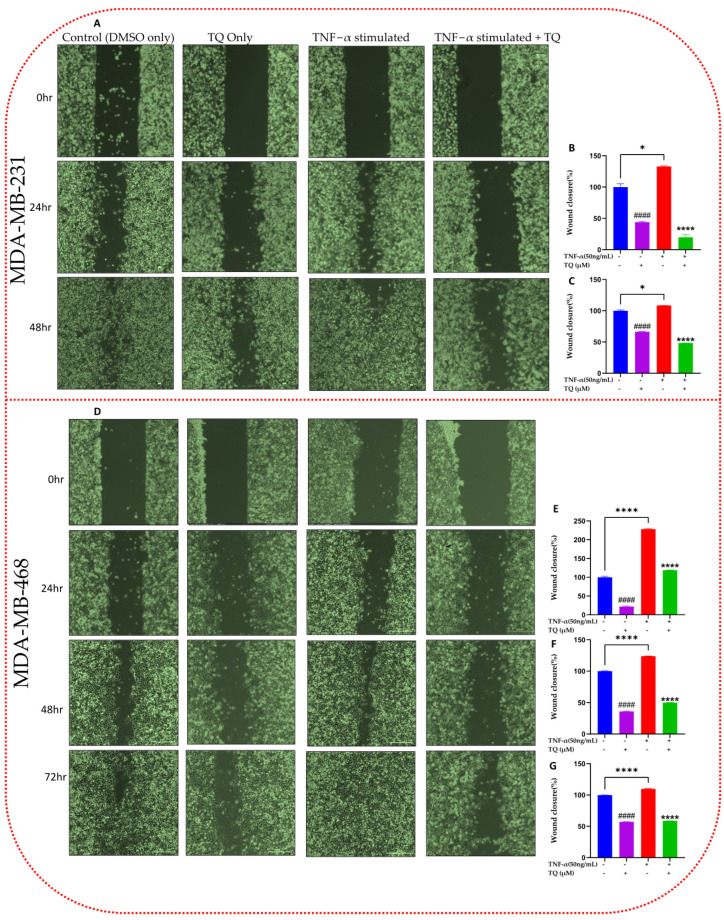
Effect of TQ on the migration assay in TNF-α-stimulated MDA-MB-231 (**A**) and MDA-MB-468 (**D**) cells: TNBC cells of the same cell densities (3.5 × 10^4^ /mL) were seeded in 2-well inserts and incubated overnight. Following removal of the insert and wash with 1× PBS after 24 h, the cells were treated with TNF-α (50 ng/mL) ± TQ (10 µM) and imaged at 0 time (phase contrast, 20×). Both cells were incubated until the TNF-α-only stimulated cell gaps were closed. At 24 and 48 h, cells were again imaged for both cells, and after 72 h for MDA-MB-468 cells; then, the gap width was analyzed. The generated data (**B**,**C**,**E**–**G**) are presented as the mean ± SEM of three independent experiments in the bar graph. An unpaired *t*-test was used to analyze the significance of the difference between TNF-α-stimulated vs. < 0.1 DMSO-treated nonstimulated cells (#) and TNF-α-activated vs. TQ cotreated cells (*). * *p* < 0.05 and ####/**** *p* < 0.0001. Scale bar: 300 µm.

**Figure 5 ijms-24-09878-f005:**
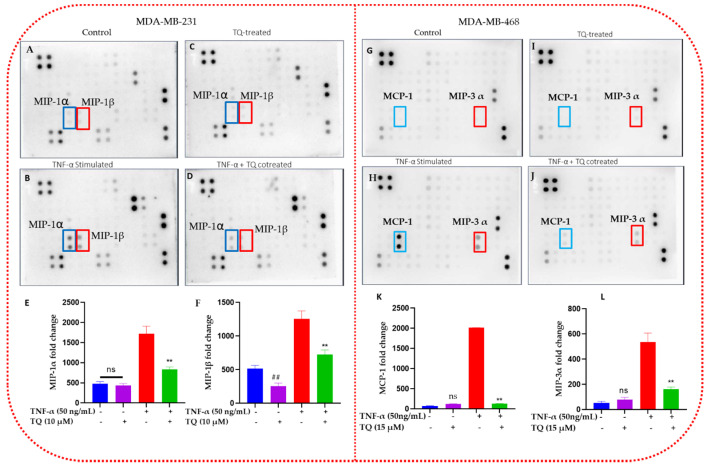
This figure shows how TQ inhibits chemokine in TNBC cells activated by TNF-α: The expression of several chemokines/cytokines was assessed using human cytokine membrane array blots in the cell-free supernatant. The four blots represent control, TQ-treated, TNF-α-stimulated, and cotreated (TNF-α + TQ cells) TNBC cells. The photographs caught the MCP-1 (CCL2), MIP-1α (CCL3), MIP-1β (CCL4), and MIP-3α (CCL20) chemokine’s changing expressions. The affected chemokine in MDA-MB-231 cells MIP-1α (CCL3) and MIP-1β (CCL4) are indicated by a blue and red frame on the human cytokine microarray map, respectively (**A**–**D**). The affected chemokine in MDA-MB-468 cells MCP-1 (CCL2) and MIP-3α (CCL20) is indicated by a blue and red frame on the human cytokine microarray map, respectively (**G**–**J**). (**E**,**F**,**K**,**L**) shows the measuring amount of extracellular CCL3, CCL4, CCL2, and CCL20 that TNBC cells with various treatments release, respectively. An unpaired *t*-test was used to evaluate whether there was a statistically significant difference between TQ-treated cells and control cells (#) or between TNF-α-stimulated cells and cotreated cells (*). ns: not significant. **/## *p* < 0.01.

**Figure 6 ijms-24-09878-f006:**
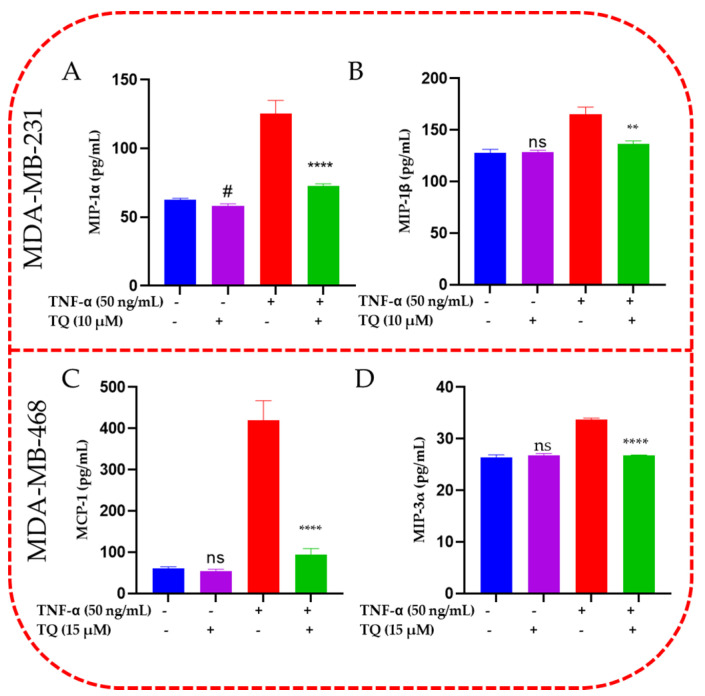
ELISA quantifications and validation for MIP-1 alpha, MIP-1 beta, MCP-1 and MIP-3 alpha release from TNF-α-stimulated MDA-MB-231 (**A**,**B**) and MDA-MB-468 cells (**C**,**D**). The normalized data show the protein expression (pg/mL) in four samples of cell supernatants, namely control, TQ-treated, TNF-α-treated cells, and cotreated cells (TNF-α + TQ). The data generated from three independent experiments are presented as the mean ± SEM. The significant difference between TQ-treated cells vs. control cells (#) or between TNF-α-stimulated vs. cotreated cells (#) groups was analyzed using an unpaired *t*-test. # *p* < 0.05, ** *p* < 0.01, and **** *p* < 0.0001 are considered significant differences, ns not significant.

**Figure 7 ijms-24-09878-f007:**
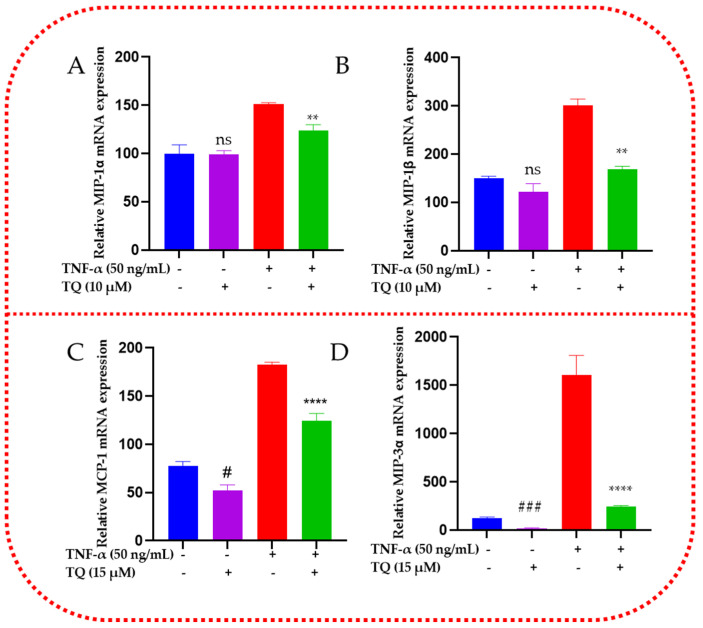
Effect of TQ on the chemokines in TNF-α-stimulated MDA-MB-231 (**A**,**B**) and MDA-MB-468 (**C**,**D**) cells: GAPDH-normalized PCR data revealed a significant upregulation in CCLE, CCL3, CCL4, and CCL20 genes in TNF-α-stimulated TNBC cells. These genes were significantly repressed in 10 µM or 15 µM TQ in MDA-MB-231 and MDA-MB-468 cells. The data are presented as the mean ± SEM of three independent studies. An unpaired *t*-test was used to analyze the significance of the difference between TQ vs. nonstimulated cells (#) and TNF-α-activated vs. cotreated cells (*). # *p* < 0.05, ** *p* < 0.01, ### *p* < 0.001, and **** *p* < 0.0001; ns: nonsignificant.

**Figure 8 ijms-24-09878-f008:**
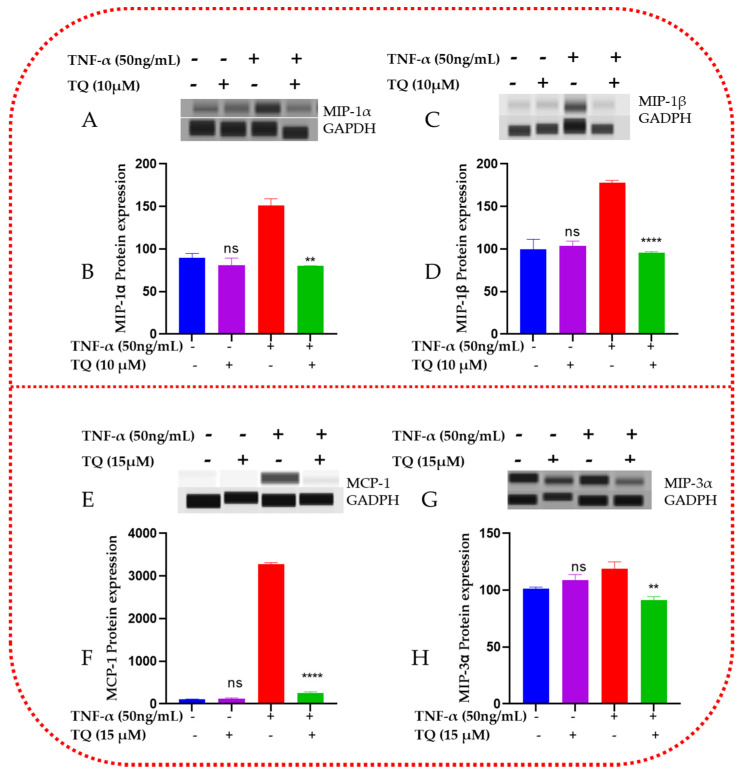
Effect of TQ on CCL2, CCL3, CCL4, and CCL20 protein expression in TNBC cells stimulated by TNF-α: For each experiment, four samples representing the resting, TQ-treated (10 µM in MDA-MB-231 cells and 15 µM in MDA-MB-468 cells), 50 ng/mL TNF-α-stimulated, and cotreated (TNF-α + TQ) cells were used to evaluate the expression of various proteins. Band immunodetection for MDA-MB-231 (**A**,**C**) and MDA-MB-468 (**E**,**G**) cells was carried out using the automated, straightforward Western system with its Compass software. GAPDH-normalized data for MDA-MB-231 (**B**,**D**) and MDA-MB-468 (**F**,**H**) cells were quantitatively analyzed. An unpaired *t*-test was used to determine the significance of the difference between TNF-treated cells vs. control cells (*) or between TNF-α-stimulated cells vs. cotreated cells (#) groups. Significant differences are defined as ** *p* < 0.01 and **** *p* < 0.0001; ns: not significant.

## Data Availability

All data generated or analyzed during this study are included in this published article.

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
