# Peer review of "Thymoquinone Inhibition of Chemokines in TNF-α-Induced Inflammatory and Metastatic Effects in Triple-Negative Breast Cancer Cells"

_ijms, 2023, doi:10.3390/ijms24129878_

Round 1
Reviewer 1 Report
The authors present results supporting action of TQ on TNF-a induced triple-negative breast cancer cells.
Please be clear about the procedence of TQ, The number and purity declared in methods (MKCC0600) is not the same which that available in SIGMA-Aldrich page. It was from this company? There is information about the source of this used reagent?
The compilation of mechanisms of action reported for TQ in TNBC cells is desirable in the discussion.
In conclusions should be clear sentencing this compound could be found in nature, since that used in these assays is not clearly a natural compound.
Author Response
Reviewer 1
The authors present results supporting the action of TQ on TNF-a-induced triple-negative breast cancer cells.
Please be clear about the precedence of TQ, The number and purity declared in methods (MKCC0600) are not the same as that available on the SIGMA-Aldrich page. It was from this company. There is information about the source of this used reagent.
- Response: corrected as suggested as "purity > 98%, cat # 27466-5G, Lot# MKCC0600" on page 13, Line 428.
The compilation of mechanisms of action reported for TQ in TNBC cells is desirable in the discussion.
- Response: Included as suggested shown on page 11, lines 324-333 as "TNBC and other types of tumor cells have shown anticancer activity in response to the refined natural compound TQ. Both in vitro and in vivo settings have been used to study TQ's anticancer properties. Among the major anticancer mechanisms of TQ, according to a thorough review of the literature, including cell cycle arrest, induce apoptosis by activating different caspases, including caspase-8 and 9, inhibit protein expression of anti-apoptotic genes such as Bcl-2, Bcl-Xl XIAP, and surviving, lower AKT phosphorylation, induce p38 phosphorylation, and, as well as interfere with the survival of cells involving the PI3-K/Akt pathway. The anticancer effects of TQ on the MDA-MB-231 and MDA-MB-468 cell lines have recently been reported. These effects include cell cycle arrest, induction of apoptosis, regulation of numerous apoptosis-regulated genes, inhibition of proliferation, antagonism of clonogenicity, and inhibition of metastasis".
In conclusion, it should be clear sentencing that this compound could be found in nature since that used in these assays is not clearly a natural compound.
- Response: Modified as suggested shown on page 17, lines 642-645 as "In conclusion, the results of this study indicate that genetically different cell lines may respond to the natural compound TQ in various ways, as TQ targets CCL3 and CCL4 in MDA−MB−231 cells and CCL2 and CCL20 in MDA−MB−468 cells.
Reviewer 2 Report
Comments and Suggestions for Authors:
Title: Thymoquinone Inhibition of the Inflammatory and Metastatic Effects of TNF−α−induced Chemokines in Triple-Negative Breast Cancer Cells
Through a series of experiments, the author pointed out that TQ, a natural product with antioxidant and anti-tumor activities, can inhibit the proliferation, migration, and expression of some cytokines in TNF-alpha induced MDA−MB−231 and MDA−MB−468 cells. It is an important study, but there are some problems, which must be solved before it is considered for publication.
1- In the introduction, more attention should be paid to the known effects of Thymoquinone.
2- In Figure 4, the author should indicate the treatment conditions for each image. According to my understanding, the results in Figure 4 show that Thymoquinone treatment can promote the migration of TNF-alpha induced MDA−MB−231 and MDA−MB−468 cells. This is clearly inconsistent with the conclusion.
3- Authors should check the spellings in current manuscript, such as line 59-62 on page 2, line 108 on page3, line 181 on page 5 and line 448 on page 14.
There are large number of spelling errors in the current manuscript, which must be corrected before it is considered for publication.
Author Response
Reviewer 2
Through a series of experiments, the author pointed out that TQ, a natural product with antioxidant and anti-tumor activities, can inhibit the proliferation, migration, and expression of some cytokines in TNF-alpha-induced MDA−MB−231 and MDA−MB−468 cells. It is a critical study, but some problems must be solved before it is considered for publication.
- In the introduction, more attention should be paid to the known effects of Thymoquinone.
- Response: Corrected as suggested, included on pages 2 and 3, lines 86-97. TQ has been investigated for its antioxidant, anticancer, and anti-inflammatory activities in both in vitro and in vivo settings. Numerous studies have recently shown that TQ has anticancer properties against several cancer types, including breast, lung, laryngeal, ovarian, osteosarcoma, and myeloblastic leukemia. Among the significant anticancer mechanisms of TQ, according to a thorough review of the literature including cell cycle arrest, induce apoptosis by activating different caspases, including caspase-8 and 9, inhibit protein expression of anti-apoptotic genes such as Bcl-2, Bcl-Xl XIAP, and surviving lower AKT phosphorylation, induce p38 phosphorylation, and, as well as interfere with the survival of cells involving the PI3-K/Akt pathway. Recently we have reported the anticancer effects of TQ on MDA−MB−231 and MDA−MB−468 cell lines, including cell cycle arrest, apoptosis induction and regulating of several apoptosis-regulated genes, anti-proliferation, anti-clonogenicity, and anti-metastasis.
- In Figure 4, the author should indicate the treatment conditions for each image. According to my understanding, the results in Figure 4 show that Thymoquinone treatment can promote the migration of TNF-alpha-induced MDA−MB−231 and MDA−MB−468 cells. This is clearly inconsistent with the conclusion.
- Response: Figure 4 is Corrected as suggested. The treatment conditions were included in each image, and the sequence of the image was modified as that of the corresponding bar graphs shown on page 6.
- Authors should check the spelling in the current manuscript, such as lines 59-62 on page 2, line 108 on page 3, line 181 on page 5, and line 448 on page 14.
- Response: Checked and corrected as suggested, included on page 2, lines 59-60; on page 3, lines 116-118; on page 5, lines 187-193; and on page 14, lines 459-460.
- There are a large number of spelling errors in the current manuscript, which must be corrected before it is considered for publication.
- Response: Checked all spelling errors and corrected throughout the text

Reviewer 3 Report
The paper named “Thymoquinone Inhibition of the Inflammatory and Metastatic Effects of TNF−α−induced Chemokines in Triple-Negative Breast Cancer Cells” utilized TNF-stimulated TNBC cells to examine cascade of events involving cytokines, CCs, adhesions, MMPs, and angiogenesis in order to evaluate the anti-inflammatory and antimetastatic effects of the natural product thymoquinone (TQ). The study is well done and the results showed that TQ inhibits CCL2, CCL3, CCL4, and CCL20 transcription and reduces the corresponding proteins.
Only some minor questions are required
1) In line 138 author states that the variations on TQ´s reaction between cells may be due to several molecular processes. What processes are referred to?
2) In figure 2 is very curiously that in MDA-MB-231 author do not have variation between replicates since 20 mM of TQ.
3) Between line 164-165 author say that “The results further supported the same responses of TNF−α stimulated TNBC cells to TQ treatment” However in figure 2 differences between cells can be seen. In fact it is possible to say that in MB-235 TQ has a more pronounced effect.
4) In figure 3B again although in this case in MDA-MB-468 cells after 10mM of TQ author do not have variations between replicates.
5) Figure 4 is confusing the treatment must be included in the figure. It seems that TQ alone induces migration, please can explain this issue?
6) Author analyzed cytotoxicity, and proliferation suing in both cases Alamar blue. I think that using FACS it is possible obtain more information because it is possible known what is the mechanism of cytotoxicity (apoptosis...) or only cell cycle stop.
Author Response
Reviewer 3
- In line 138 author states that the variations in TQ's reaction between cells may be due to several molecular processes. What processes are referred to?
- Response: Corrected as "These effects' variability may indicate the different anti-TNBC mechanisms of TQ in these two genetically distinct cell lines," included on page 4, lines 144-140.
- In figure 2 is very curious that in MDA-MB-231 author do not have variation between replicates since 20 mM of TQ.
- Response: Corrected as suggested as "Compared to MDA-MB-468 cells, the antiproliferative effect of TQ on MDA-MB-231 cells exhibits a similar response pattern starting at 20 µM regardless of exposure time or concentration "included on page 4, lines 140-141.
- Between line 164-165 author say that "The results further supported the same responses of TNF−α stimulated TNBC cells to TQ treatment" However, in Figure 2, differences between cells can be seen. In fact, it is possible to say that in MB-235, TQ has a more pronounced effect.
- Response: Corrected as suggested, included on page 4, lines 166-166, as the findings further confirmed that TNF−α triggered TNBC cells respond similarly to TQ treatment. TNF−α stimulated MDA-MB-231 TNBC cells responding to TQ less sensitively against clonogenicity than TNF−α stimulated MDA-MB-468 TNBC cells".
- In Figure 3B, again, although in this case in MDA-MB-468 cells after 10mM of TQ author does not have variations between replicates.
- Response: Modified as suggested, included on pages 4 and 5, lines 163-169 as" In contrast, in TNF−α stimulated MDA−MB−468 TNBC cells at 5 µM of TQ, the colony formation was reduced by 80%, and at 10 µM and onwards, irrespective of time exposure and concentration, the colony formation is reduced by more than 90% (p<0.0001) (Figure 3B). The findings further confirmed that TNF−α triggered TNBC cells respond similarly to TQ treatment, with MDA-MB-231 TNBC cells responding to TQ less sensitively against clonogenicity than MDA-MB-468 TNBC cells".
- Figure 4 is confusing; the treatment must be included in the figure. TQ alone induces migration. Please can you explain this issue?
- Response: Figure 4 is Corrected as suggested. The treatment conditions were included in each image, and the image sequence was modified as that of the corresponding bar graphs shown on page 6.
- The author analyzed cytotoxicity and proliferation suing in both cases Alamar blue. I think that by using FACS, it is possible to obtain more information because it is possible to know what the mechanism of cytotoxicity is (apoptosis...) or if only the cell cycle
- Response: Even though, as the reviewer pointed out, FACS may be used to measure the same measurement parameters. We decided to test both cell lines' clonogenicity as well as cell viability and proliferation using Alamar Blue.

Round 2
Reviewer 2 Report
Authors addressed all the comments and improved the manuscript.